# Clays as Vehicles for Drug Photostability

**DOI:** 10.3390/pharmaceutics14040796

**Published:** 2022-04-05

**Authors:** Monsuêto C. da Rocha, Thais Galdino, Pollyana Trigueiro, Luzia M. C. Honorio, Raquel de Melo Barbosa, Santiago M. Carrasco, Edson C. Silva-Filho, Josy A. Osajima, César Viseras

**Affiliations:** 1Interdisciplinary Laboratory for Advanced Materials, LIMAV, UFPI, Teresina 64049-550, PI, Brazil; monsueto07@hotmail.com (M.C.d.R.); thais.g.andrade@hotmail.com (T.G.); pollyanatrigueiro@gmail.com (P.T.); luzia_quimica@yahoo.com.br (L.M.C.H.); edsonfilho@ufpi.edu.br (E.C.S.-F.); 2Materials Science and Engineering Postgraduate Program, PGCM/CCSST, UFMA, Imperatriz 65900-410, MA, Brazil; 3Department of Pharmacy and Pharmaceutical Technology, Faculty of Pharmacy, Campus de Cartuja s/n, University of Granada, 18071 Granada, Spain; cviseras@ugr.es; 4X-ray Laboratory, Center for Research, Technology and Innovation, University of Sevilla, CITIUS, 41012 Sevilla, Spain; sanmedi@us.es

**Keywords:** clay minerals, pharmaceuticals, hybrids, photosensitivity, stability, biotechnological application

## Abstract

Clay minerals are often used due to their high adsorption capacity, which has sparked interest in their biological applications to stabilize drugs and pharmaceutical products. This research aims to summarize information about the stability of drugs, cosmetics, dermocosmetics, and pharmaceutical compounds incorporated in the structure of different clay minerals. The databases used to search the articles were Web of Science, Scopus, PubMed, and Science Direct. Photostabilization of these compounds is reviewed and its importance demonstrated. For biological applications, the increase in solubility and bioavailability of clay minerals has proven useful for them as drug carriers. While their natural abundance, low toxicity, and accessible cost have contributed to classical applications of clay minerals, a wide range of interesting new applications may be facilitated, mainly through incorporating different organic molecules. The search for new functional materials is promising to challenge research on clay minerals in biological or biotechnological approaches.

## 1. Introduction

The same cosmetic or pharmaceutical products can be applied in three ways—dermal, oral, or ocular. Products used for dermal application must have good adhesion to the skin; stability; and especially not cause any type of allergy, toxicity, or phototoxicity. Some formulations can increase the stability of bioactive compounds and clay minerals are currently a trend in the cosmetics and dermocosmetics industries, especially when consumers demand products of natural origin [1,2].

Drug molecules, especially carbonyl, nitro, alkene, aryl chloride, or phenolic groups, are very susceptible to absorbing the wavelengths of light associated with sunlight, degrading the drug due to oxidation, and reducing the period that the product retains good stability. As a result of these changes, the product reduces the dose of active pharmaceutical ingredients. New potentially toxic degradation products form, including chemical species with unpaired electron free radicals. These changes may occur during the distribution stages of the pharmaceutical products, reducing their shelf life, even if the packaging of the products and stability studies during product development indicate that they are very rare. Photostability studies, carried out at various stages of the product development process, investigate the effect of light on drugs or cosmetic actives and formulated products.

The photostability studies use artificial-daylight fluorescent lamps that emit long-wavelength ultraviolet and visible light to simulate indirect, indoor sunlight at a controlled temperature (usually 25 °C). Reduction of the drug molecule, formation of photodegradation products, or color changes are determined, and light protection of the product is designed following the obtained results. Therefore, studies are designed and carried out to prevent degradation (under indoor light) before being administered, but sunlight on the product, once issued, is not usually studied.

However, once administered, and mainly in topical dosage forms, photodegradation can occur in most cases, producing free radicals that interact with surrounding skin tissues and can cause severe effects. Photodamage can occur to the human body in response to exposure to ultraviolet radiation (UV), divided into UVA (320–400 nm), UVB (280–320 nm), and UVC (200–280 nm). UVA directly penetrates the earth’s surface, and UVB radiation does too but to a lesser extent. UV radiation can cause extreme skin sensitivity when exposed to sunlight induced by topical application of pharmaceuticals [3,4]. These photoallergic and phototoxic responses may result from skin interaction with pharmaceuticals that degrade upon light exposure [5]. Phototoxicity can cause direct damage to skin tissues (such as pain, redness, or inflammation) from a single exposure to sunlight. Photoallergies are delayed responses in which inflammatory reactions (such as redness, scaling, or itching) due to the application of the substance will only be initiated with subsequent exposure to sunlight [6,7].

The study of photostability and photoprotection mechanisms for these compounds has become a topic of growing interest mainly due to the increased consumption of pharmaceutical products. Therefore, strategies have been developed to increase the stability of drugs. For example, Lai et al. [8] used a suspension formulation to increase the photostability of tretinoin, which is also known as retinoic acid. Tretinoin is derived from vitamin A and is widely used to lighten skin blemishes and treat acne. The authors noted that the application of nanosuspension in topical tretinoin delivery increases the photostability of the drug. Raffin et al. [9] investigated the photostability of pantoprazole in polymer microparticles. Pantoprazole sodium is a drug indicated for cases of mild gastrointestinal discomfort and to treat mild gastric lesions. The authors verified the increase in stability under UVC irradiation of microencapsulated pantoprazole with different polymers. Isradipine presents low solubility, instability under light, and short elimination half-life [10]. In addition, isradipine is a calcium channel blocker and is indicated for blood pressure control. In the studies by Park et al. [10], the complexation between isradipine and β-cyclodextrin increased its solubility and photostability. Tiwari et al. [11] showed that itraconazole loaded in essential oil microemulsions enhanced photostability under UV and visible light irradiation. Itraconazole, an imidazole derivative, is a very effective drug for treating severe fungal infections. In addition, photostability results indicated that microemulsions had a protective effect against drug decomposition.

Natural raw materials have become increasingly sought after for application in the pharmaceutical and cosmetic industries. Clay minerals have interesting technological characteristics that promote efficient results in specific product functions, mainly in the stabilization of emulsions and dispersions of formulations [12]. Clay minerals are extensively used due to their properties, such as high specific surface area, optimal rheological characteristics, and excellent adsorption capacity [13]. Accordingly, clay minerals have favorable chemical and structural features as photoprotective materials for different organic molecules [14,15]. Incorporating drugs in different clay minerals has received much attention for biological applications [16,17]. The orientation and stabilization of organic molecules confined in the interlamellar spacing of some clay minerals result in specific structures depending on the cation exchange capacity and the amount of drug incorporated [18,19]. Drug–clay mineral hybrids can improve drug solubility [20], increase bioavailability [21,22], modify/control release [23,24], and increase stability [25,26].The databases of articles used were Web of Science (Topic—TS), Scopus, PubMed (All Fields), and Science Direct (Find articles with these terms). Clay minerals and drugs was the main topic, and next the refinement keywords were delimited: clays and pharmaceutics, clays and cosmetics, clays and dermocosmetics, drug stability, clays and drug stability, drug photostability, clays and drug photostability, clays and biological applications, and clays and biotechnological applications.

This work aims to gather information about the photochemical behavior of drugs and pharmaceutical products before and after incorporation into clay minerals. The role of clay minerals in the photostabilization of these compounds will be evaluated. In addition, this review not only summarizes studies about drug stabilization in the structure of different clay minerals but also presents other perspectives about drug–clay mineral hybrids in terms of biotechnology concern.

## 2. Structure of Clays and Clay Minerals: Basic Concepts

Clay minerals are hydrated aluminum phyllosilicates, and their classification mainly depends on their type of structure and chemical composition. The possible arrangements of tetrahedral (T) and octahedral (O) sheets classify different groups of clay minerals. As shown in Figure 1, the basic units of the phyllosilicates can form the 1:1 and 2:1 combinations. The TO-type structure (1:1) has an octahedral aluminum sheet bonded to a tetrahedral silicon sheet. The chemical structural formula of the unit cell is [Si_2_Al_2_O_5_(OH)_4_], while the structure of the TOT type (2:1) consists of an octahedral sheet Al^3+^, Fe^3+^, or Mg^2+^ sandwiched between two tetrahedral sheets of Si^4+^ or Al^3+^. The chemical formula of the unit cell is [Si_8_Al_4_O_20_(OH)_4_] or [Si_4_Al_2_O_10_(OH)_2_] for a half unit cell. Isomorphic substitutions promote the formation of a negative lamellar charge, which is compensated for by the presence of interlamellar cations. The porosity of clay minerals can be characterized by interlamellar spacing, which is generally microporous and has interparticle pores, which can be mesoporous or macroporous [27,28,29]. Of the various types of clay minerals found, we will review some specifics used in selected works in the literature.

Bentonite is a natural clay constituted predominantly of ≥50% smectite phase, commonly montmorillonite, and is widely used in studies on the adsorption of different organic molecules [30,31].

Smectites are 2:1 phyllosilicates with a total (negative) layer charge between 0.2 and 0.6 per half unit cell with marked lamellar structure [28]. Montmorillonite (Mont) is a layer of dioctahedral smectite composed of two continuous silicon tetrahedral sheets and an aluminum octahedral sheet (2:1 type). A negatively charged layer arises from the substitution of Al^3+^ for Mg^2+^ and other cations in octahedral sites, presenting Na^+^ or Ca^2+^ as interlayered cations [32,33]. Montmorillonite can interact with biomolecules in different ways [34]. Saponite (Sap) clay mineral belongs to the naturally occurring trioctahedral smectite and has layers of Si^4+^ tetrahedron and Mg^2+^ octahedron. Various functional cations can exchange the interlayered cations for potential applications in catalysis and adsorption [35]. Hectorite (Hec) is a natural white clay mineral where the octahedral sites are occupied with Mg^2+^ or Li^+^ cations, presenting an empirical formula of Na_0.3_(Mg,Li)_3_ Si_4_O_10_(OH)_2_ [36]. Laponite (Lap) is a synthetic clay that shares structural similarities with naturally occurring hectorite. Both clay minerals belong to the smectite group of phyllosilicates, with one octahedral magnesium sheet sandwiched between two tetrahedral silica sheets. Laponite has a chemical formula of Na^+0.7^[(Si_8_Mg_5.5_Li_0.3_)O_20_(OH)_4_]^−0.7^, in which some of the magnesium is replaced by lithium cations [37].

Vermiculite (Ver) is a natural trioctahedral phyllosilicate with a structure consisting of TOT layers composed of two tetrahedral silica sheets attached to a central magnesium octahedral sheet. The negative layer charge arises from the isomorphic substitution in tetrahedral and octahedral sheets, and it is balanced by the presence of different exchangeable interlayered cations [38,39].

Palygorskite (Pal) (SiMg_8_O_20_(OH)_2_(H_2_O)·4H_2_O is a kind of hydrous magnesium-aluminosilicate clay mineral with a chain-lamellar structure, and its crystal structure is rod-like, fiber-like, or fiber aggregate [40]. Negative charges are formed from substitutions of Si^4+^ in the tetrahedral sheet by Al^3+^. Palygorskite clay mineral could be used as a drug carrier or catalytic support [41].

Sepiolite (Sep) is a hydrated magnesium silicate with a typical structural formula of Si_12_O_30_Mg_8_(OH)_4_(OH_2_)_4_ 8H_2_O with a fibrous morphology and intercrystalline channels. Sepiolite fibers exhibit a high specific surface area, microporosity, and good chemical stability and appear to be an attractive support material for drug adsorption and release [42,43].

Kaolinite (K) [Si_2_Al_2_O_5_ (OH)_4_] is a phyllosilicate from kaolin. The layers that make up the kaolinite structure are formed by a silicon tetrahedron sheet bonded to an aluminum octahedron sheet forming a 1:1 lamellar structure [44]. Kaolinite has already been investigated for the treatment of cancer in different drug-loaded formulations [45].

Halloysite nanotubes (HNTs) consist of thin tubular morphology with the chemical composition Al_2_(OH)_4_Si_2_O_5_(2H_2_O). Halloysite is a 1:1 clay mineral, meaning that each layer is composed of silica tetrahedral and aluminum octahedral sheets. The advantages of HNTs include biocompatibility, low cost, hydrophilic properties, and the capability to encapsulate drugs [46,47].

Layered double hydroxides (LDHs) are called “anionic clays” because of their anion-exchange properties. As hydrotalcite (HTlc) is one of the most representative minerals of the group, LDHs are also called “hydrotalcite-like compounds” (HTlc) [28]. LDHs are lamellar materials based on stacking positively charged sheets. Hydrated anions are located in the interlayered space to maintain global neutrality. LDH-based hybrid materials can be used in biosensors, biocatalysis, and drug delivery [48].

## 3. Approaches on the Stability of Pharmaceutical-Clay Minerals Systems

The most common paths of drug degradation are thermolytic, oxidative, and photolytic processes. Knowledge about the degradation mechanisms of these compounds is crucial for development of more chemically stable products [49]. These substances can degrade due to pH variation, temperature, or light exposure. Photodecomposition can decrease the potential of the product, therapeutic inactivity, or the formation of by-products during the storage and manipulation of the drug. Therefore, studies about the chemical/photochemical stability of drugs and pharmaceutical products can be essential to estimate (i) production, packaging, and finishing steps; (ii) drug pharmacodynamics; and (iii) development of new systems [50].

Different clay minerals can be used in conventional pharmaceutical dosage forms. Kaolinite, smectite, and fibrous clay minerals are widely used in stabilization studies and control the release of drugs. Smectites have been the most commonly used substrates as they can retain large amounts of the drug due to their high cation exchange capacity. HNTs and LDHs can also be used. New clay minerals and derived materials are continually being explored, and as they advance, new pharmaceutical formulations are constantly being developed to achieve the ideal therapeutic effects and pharmacological action [13,51,52]. Examples of the main clay minerals applied for drug stabilization by the cosmetic and pharmaceutical industry are presented in Table 1.

### 3.1. What Is the Role of Clay Minerals in the Photostability of Drugs?

The photostability of a drug can be defined as its response to exposure to sunlight, UV, or visible light that leads to color, purity, or quality variation as a function of time [61,62]. Different approaches can control the photodegradation of drugs and increase their photostability, as previously described. Clay minerals are attractive materials for drug photoprotection due to their striking features and structural modification capacity.

Different strategies to improve the photostability of drugs have been demonstrated [4,63]. In addition, some studies have tested the ability of different formulations to increase the stability of organic compounds [64]. However, there are still few publications on clay minerals as photoprotective matrices in the drug–clay mineral system. Light exposure promotes the formation of reactive oxygen species that degrade some compounds by photochemical reactions. Therefore, the encapsulation of organic molecules in the interlamellar spacing of certain clay minerals appears to reduce or prevent this photo-oxidative degradation [65]. Ambrogi et al. [65] investigated the role of montmorillonite and a halloysite in the photostability of piroxicam, which is a non-steroidal anti-inflammatory drug that induces cutaneous reactions in patients after exposure to sunlight. Therefore, piroxicam is photosensitive and can cause photoallergic reactions. The authors prepared the drug–clay mineral nanohybrids by the method of physical mixing and by intercalation in solution. The photostability studies were performed using a Xenon lamp as the light source, and the powder sample was irradiated for 120 min. The results indicated that the new hybrids were obtained with many drugs incorporated into them. From the irradiation experiments, the authors concluded that the montmorillonite-based hybrids showed more photostability than the halloysite-based hybrids [65]. The strong interactions between the drug and the surface of montmorillonite increased the photostability of piroxicam.

In another study, Ambrogi et al. [66] evaluated the protective effect of promethazine (PRO) intercalated with montmorillonite. PRO is a photolabile phenothiazine drug with antihistaminic activity, used by systemic administration to treat nausea, vomiting, and motion sickness and topically for insect bites, erythema, and itching. The authors prepared the PRO-montmorillonite hybrids and their formulations. They also performed release tests from the formulations. For the photostability tests, experiments were conducted with irradiation under UV light, in solution for 60 min, and in powder form with an exposure period of 120 min. Based on the release experiments, the authors concluded that the hybrids have a controlled diffusion mechanism. Furthermore, PRO intercalated in the inorganic matrix had a decreased degradation rate, indicating photo stabilization effects based on the irradiation tests.

Marques et al. [67] presented a discussion on the use of montmorillonite (K10) and MCM-41 as substrates for incorporation of nifedipine and carried out controlled release and light stability experiments. Nifedipine, a potent calcium channel dihydropyridine blocking agent, is widely used to treat hypertension and as an anti-anginal agent and has low bioavailability after oral administration due to its low solubility in water. The drug was incorporated by intercalation in solution. The drug stability tests were performed in two ways: (i) the samples were exposed to indirect sunlight for one month; and (ii) the samples were kept in a closed container, in the dark conditions, with a controlled atmosphere for 18 months. The authors observed the intercalation of nifedipine in montmorillonite (as shown in Figure 2), whereas nifedipine occupied the pores into MCM-41 structure. MCM-41 presents higher availability than found for the K10 clay mineral in release tests. All hybrids exhibited enhanced photo stabilization of the drug after immobilization inside the host materials from the stability experiments. The hybrid formulation materials also work as a stabilizing agent for drug storage [67].

A nanocomposite of halloysite nanotubes complexed with microcrystalline cellulose was used to incorporate nifedipine in the studies of Yendluri et al. 2017 [68]. The authors investigated the release kinetics and photostability of the material in tablet form. The tablets were prepared with 50 wt% of the drug-loaded halloysite mixed with 45 wt% of biopolymers. Photodegradation of nifedipine-loaded halloysite structure was evaluated in a reactor with Xenon lamps corresponding to 30 K Lux and 60 K Lux as the light source, as shown in Figure 3. Samples were irradiated for more than 2000 h. The results demonstrated that halloysite is a functional excipient for formulations for the controlled release of drugs with high photoprotective power.

Tetracycline is one of the most used antibiotics because of its low cost, broad-spectrum, and antimicrobial properties. However, tetracycline can absorb light and cause photosensitivity due to the formation of photoproducts after exposure to light [69]. A study by da Rocha et al. [69] evaluated the photoprotection potential of tetracycline incorporated in different clay minerals. Tetracycline in an aqueous solution was incorporated into montmorillonite, sepiolite, and palygorskite. The photocatalytic tests placed 2 g of each hybrid on scattered glass plates to obtain a material layer with thickness below 1–2 mm. The samples were exposed to a mercury vapor lamp as a source of UV radiation up to 200 h. From the thermal analysis results, high incorporations of tetracycline and higher thermal stability in the structures of the studied clay minerals were observed. Furthermore, photostability tests indicated that the tetracycline adsorbed into clay structures was not easily degraded by UV irradiation, especially in the lamellar clay minerals.

### 3.2. Strategies for Cosmetic/Dermocosmetic Stability

Therapeutic effect and mechanism of action can be determined by the chemical composition and physicochemical properties of clay minerals when used as active ingredients in cosmetic formulations and pharmaceutical products. Thereby, clay minerals can be used in dermatological protectors, cosmetic creams, and emulsions. These applications are mainly due to its purity, texture, particle size, abrasiveness, non-toxicity, opacity, astringency, and heat-retention capacity [70,71]. With the increasing demand for the consumption of cosmetic products, the use of clays and clay minerals to increase the stability and safety of active ingredients provides a fascinating study topic.

Clay minerals promote the encapsulation of drugs and active ingredients as a photoprotective coating [72], and they can act as carriers of skincare molecules for sustained delivery [73]. Based on the different interactions between organic molecules and clay minerals, new strategies and technologies are developed to increase the availability and stability of biomolecules [31,33,34,74]. Furthermore, the confinement of organic molecules in the inorganic matrix structure likely reduces photochemical oxidation, preventing the formation of toxic by-products that can lead to photoallergy or phototoxicity like skin responses in cosmetic products. Consequently, increasing the stability of the products increases their safety and effectiveness in more complex formulations [13].

Madikizela et al. [75] investigated the sun protection factor (SPF) of different formulations of cosmetic products based on natural clay minerals in the Eastern Cape Province of South Africa. The natural clay displayed a low SPF value (providing low UV protection) and cosmetic products manufactured from natural clay material have high SPF values. Therefore, they can be used for cleansing, emulsification, beautification, adsorption, and detoxification of the skin.

Nanotechnology has been widely explored to develop new cosmetic and dermocosmetic products [76]. For example, Cavallaro et al. [76] developed a protective hair treatment based on the hybrids of halloysite nanotubes and keratin to optimize the efficiency of proposed hair photoprotection coatings. Previously treated and untreated hair segments were exposed to UVA irradiation at 24, 48, and 72 h. The hair segments treated with the keratin coating maintain their protective action after three washing cycles, and this formulation increased the photoprotection of the hair structure.

Pagano et al. [77] prepared nanohybrids by combining folic acid and different layered double hydroxides, and the photoprotective properties were evaluated. Folic acid is an essential anti-aging agent widely found in product formulations available in stores and pharmacies. Toxicity results show that the formulations are safe for topical use. Intercalated folic acid has high photostability without the formation of toxic by-products.

Alavijeh et al. [78] investigated the montmorillonite used for nanoencapsulation of vitamin B6 using the cation exchange mechanism. Vitamin B6, also called pyridoxine, is an essential water-soluble biomolecule for the metabolism of amino acids. The nanoencapsulation of vitamin molecules exhibits different characteristics in response mainly to pH, and this behavior is essential in the release of these compounds in different parts of the body. The vitamin biomolecules were adsorbed in the interlayer spaces of montmorillonite, which improved its stability.

Drug–hydrotalcite nanocomposites are hybrid systems, which have characteristics suitable for use in formulations for oral or topical administration [79]. Thus, Conterosito et al. [79] used hydrotalcite for intercalating bioactive molecules. The study aims to investigate the stability of the intercalated molecules, and their influence on solubility, bioavailability, and release profile. The bioactive nanocomposites presented high chemical stability and proved to be efficient for different biological applications.

Hydrotalcite-like anionic clay was used as a substrate for Kojic [80] and ferulic acid [81]. Ambrogi et al. [80] intercalated kojic acid in hydrotalcite to increase photostability, as demonstrated in Figure 4. Kojic acid is an inhibitor of melanin synthesis used as a dermo-cosmetic agent for the treatment of melasma. A Xenon lamp performed light experiments at 330 nm, and the samples were irradiated for two hours. The results showed the Kojic molecule intercalated in hydrotalcite structure. Photostability studies revealed that organic intercalation in the inorganic matrix reduces photo-induced oxidation and addition reactions. Ferulic acid was intercalated in hydrotalcite, and its photostability was investigated in Rossi et al. [81]. Ferulic acid presented antioxidant properties and was promising as an active substance for sunscreen formulation. The solid samples were irradiated using a mercury lamp at 366 nm in photochemical studies. The authors confirmed hydrotalcite as a photoprotective agent for ferulic acid molecules; it thus has potential for sunscreen formulations.

## 4. Trends in Clays for Biotechnological Drug Stability

Biotechnology has been developing rapidly and is widely used to increase agricultural [82] and industrial yield [83]; diagnose and prevent diseases [84,85]; and advance technological innovation, including nutraceuticals [86,87], antibiotics [88], and vaccines [89,90,91]. In addition, it facilitates encoding of new enzymes and genetic modifications. However, despite significant advances in biotechnology, many new actives provided by biotechnology processes, such as amino acids, proteins, RNA, and DNA, have shallow or no stability, which is an important matter to resolve [92,93]. In addition, some factors can affect the adsorption of biomolecules (DNA, RNA, or their fragments) onto clay minerals such as types of clay minerals, molecular weight and structure, solution pH, and ionic strength [94].

For example, biofilms prepared from the intercalation of amino acids and montmorillonite had characteristics and biocompatibility in tissue engineering [95]. As amino acids are the fundamental structures that proteins construct, understanding the interactions and stabilization of different amino acids and the structures of clay minerals can lead to critical applications of these compounds related to the immune system, enzymatic catalysis, and biochemical evolution of life on earth [96,97,98,99].

Clay minerals are capable of adsorbing drug molecules [69,100], enzymes [101,102], and proteins [103,104], thus preserving the biological activity of these biomolecules and consequently being widely used in biomedicine and new insights. Encapsulating enzymes and proteins can increase their stability at different pH levels and temperatures [105,106]. These systems can be used in different science applications and therapeutic studies, generally in the biotechnology field.

Regarding prebiotic processes, the confinement of DNA or RNA molecules in clay minerals helps to elucidate the behavior of biomolecules at the origin of life [107,108]. In addition, RNA is the primary molecule for protein synthesis. Thus, several studies [109,110,111,112,113] have focused on the confinement and the interaction of RNA or its fragments (RNA homopolymers and RNA components), including nucleic acid bases of adenine (A), cytosine (C), uracil (U), thymine (T), and ribose with clay minerals, such as montmorillonite, kaolinite, and illite. The photoprotection of this system may be a crucial part of future developments in genetic engineering.

Proteins bind or adsorb to different types of clay through various physical and chemical interactions, such as cation exchange, electrostatic interactions, hydrophobic affinity, hydrogen bonding, and van der Waals forces. Johnston et al. [114] investigated the interaction of hen egg-white lysozyme with Na- and Cs-exchanged saponite by sorption, structural, and spectroscopic methods as a model system to study clay–protein interactions. Results pointed to the ion exchanges and hydrophobic interactions between saponite and lysozyme that occur in the interlayer, edges, and external surface clay mineral. On the other hand, Zhou et al. [115] studied the adsorption and desorption of the insecticidal protein (toxin) of a *Bacillus thuringiensis* strain on montmorillonite, red soil, goethite, kaolinite, and silica, which demonstrated that the interaction mechanisms involved electrostatic, hydrophobic, hydrogen bonding, and van der Waals forces. In addition, binding sites can depend on the size of molecular toxins; for smaller toxin molecules, they occur in the clay interlayer and for larger ones in the clay mineral external surface. The adsorption mechanisms of DNA and RNA onto clay minerals are similar and can involve electrostatic forces, hydrogen bonding, ligand exchange, and cation bridge, etc.—as demonstrated by [116,117,118].

Damage in the DNA structure can affect its biophysical properties and cause biological consequences [119]. Strong interactions of DNA molecules adsorbed on the surface of different clay minerals can reduce the enzymatic degradation and thermal denaturation of these biomacromolecules. Furthermore, the intercalation of DNA in lamellar clay minerals could increase their protection against cellular damage [120] and favor the abiotic synthesis of nucleic acids [121]. Consequently, the interactions between DNA molecules in the clays structures lead to a wide range of applications, including soil environment [122] and hydrogels for selective binding and molecular recognition [123].

The interactions of these bioactive molecules (proteins, DNA, and RNA) and the surface of clay minerals can expand the chemical diversity of these systems for targeting in various application areas, such as cosmetics, bioengineering, medicine, pharmaceutics, and functional foods. For example, studies published by Liu et al. [124] combined artificial microRNA (amiRNA)-mediated silencing technology and clay nanosheet-mediated delivery to prevent tomato yellow leaf curl virus, a typical monopartite begomovirus infection in tomato plants. In another work, Wen et al. [125] studied LDHs and their nanocomposites as promising candidates to (i) carry antineoplastic drugs such as camptothecin (low chemical and physical stability drug); (ii) develop nanocarriers to carry siRNA/DNA for gene therapy; (iii) develop a multi-target therapeutic vaccine; and (iv) encapsulate or immobilize photosensitizer/photothermal agents in the interlayer space for phototherapy. In general, clay minerals have a very variable chemical composition, excellent biocompatibility, high cation exchange capacity, and increased drug loading capacity, which broaden the scientific interest in searching for new materials with specific applications with different clays. However, there is still a large gap still to be fully explored and discussed.

## 5. Conclusions

The properties of clay minerals, including porous structure, specific surface area, ion exchange capacity, environmental compatibility, and selectivity, and even possible structural modifications, allow for a range of interactions with different organic molecules. Studies have demonstrated the photostability of drugs after incorporation in different clay minerals. However, the photoprotection mechanism has not yet been thoroughly elucidated. In addition to stability, these interactions allow for drug–clay mineral hybrids in different formulations aimed at various biological applications, including cosmetic and pharmaceutical products. In biotechnology and genetic engineering, interactions between biomolecules and clay minerals can play a significant role in formulating new photostable compounds. This review focused on stabilization of the different molecules onto clay mineral structures and trends for drug–clay minerals hybrids in further promising studies.

## Figures and Tables

**Figure 1 pharmaceutics-14-00796-f001:**
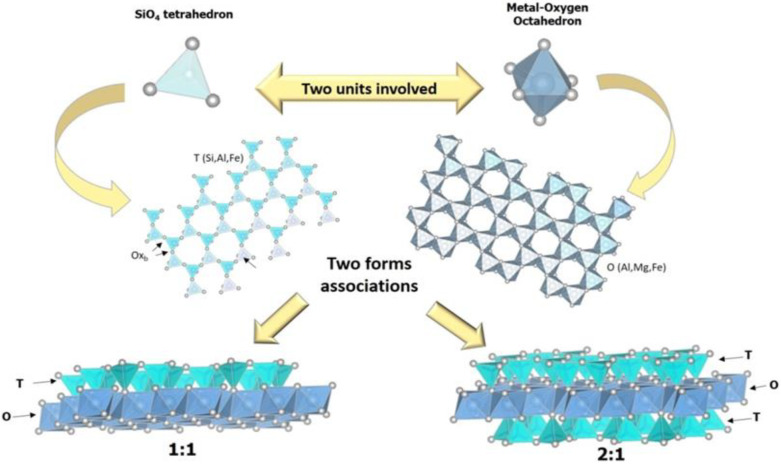
Representation of tetrahedral and octahedral sheets and 1:1 (TO) and 2:1 (TOT) arrangements in the structure of clay minerals.

**Figure 2 pharmaceutics-14-00796-f002:**
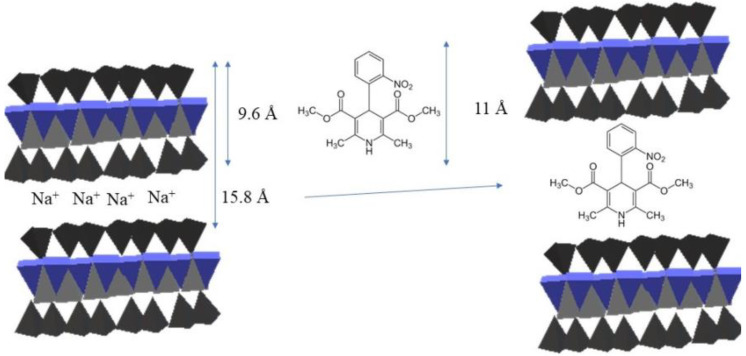
Representative scheme of nifedipine intercalation in the interlayer spacing of montmorillonite. Adapted with permission from ref. [67], 2022, Elsevier.

**Figure 3 pharmaceutics-14-00796-f003:**
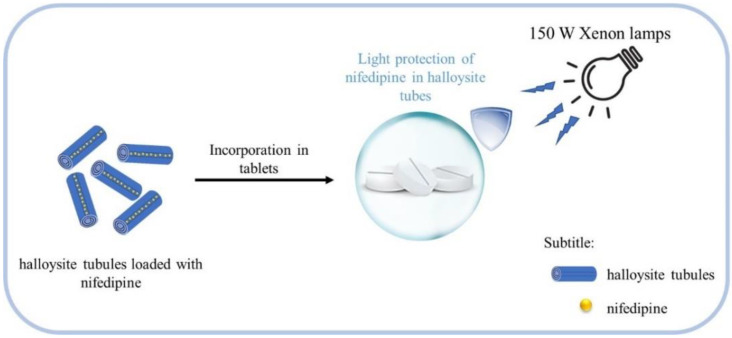
Scheme of nifedipine-loaded halloysite for study of photostability.

**Figure 4 pharmaceutics-14-00796-f004:**
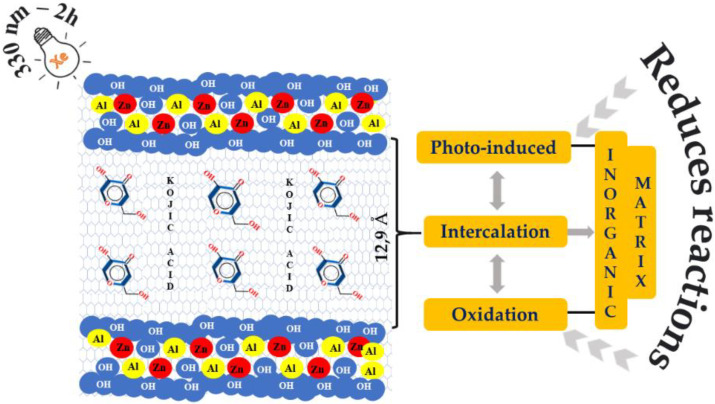
Representation of study of Kojic acid-hydrotalcite hybrid.

**Table 1 pharmaceutics-14-00796-t001:** Some clay minerals used to stabilize drug-based products in oral or topical formulations.

Clay Mineral	Function	Remarks	Ref.
Kaolinite	Excipient	Kaolinite has excellent technical properties that promote its use as excipients in oral or topical formulations with recognized efficiency to improve bioavailability and controlled drug delivery	[53]
Montmorillonite	Cosmectic ingredient	Cosmetic cream formulation was produced based on organo-montmorillonites, and biological and light irradiation tests have indicated the safety of the material for topical application	[54]
Montmorillonite	Pickering emulsions	Surfactant-free emulsions were produced by different processes. The montmorillonite-based emulsions showed good characteristics and properties to be applied as additives in cosmetic formulations	[55]
Bentonite	Formulations for cutaneous application	Pastes were prepared with bentonite modified with zinc and copper and added to phenoxyethanol (PH). The materials obtained showed excellent antimicrobial activity, allowing their use against skin infections	[56]
Palygorskite	Excipient	A pharmaceutical formulation of nifuroxazide to atalpugite was produced for application in tablets. The thermal stability and photostability of the drug encapsulated in the inorganic matrix showed promise for applications in the pharmaceutical industry	[57]
Hydrotalcite	Sunscreen formulation	Hydrotalcite intercalated with 2-phenyl-1H-benzimidazole-5-sulfonic acid was used as a matrix in a topical sunscreen formulation. The material showed high stability and photostability against drug degradation	[58]
Halloysite	Delivery of bioactive agents	Different nanoarchitectures were investigated for interactions and stabilization of drugs and biopharmaceuticals in the halloysite structure. The clay mineral has a favorable structure for different combinations with bioactive agents for anticancer therapies	[59]
LDHs	Emulsifier	LDHs are indicated as good emulsifying agents due to their physico-chemical properties and being good hosts of molecules for stabilization and protection against UV irradiation that causes skin photosensitivity	[60]

## Data Availability

Not applicable.

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
