# Peer review of "Clays as Vehicles for Drug Photostability"

_pharmaceutics, 2022, doi:10.3390/pharmaceutics14040796_

Round 1

Reviewer 1 Report

Pharmaceutics

Review "Clays as vehicles for drug stability"

  1. Which databases searched used in the review and which keywords were used
  2. It would be better if you add a table summarizing examples of a number of drugs that contain the clay and the purpose of use and if it is internal or external
  3. Line 1010: "Clays and Clay minerals structure: basic concepts". Give the sources of clays types studied, if natural or synthetic, if synthetic add their types per references.

Author Response

Dear Editor and Reviewers,

Thank you very much for your attention and for the reviewers’ comments on our manuscript ‘Clays as vehicles for drug photostability (Manuscript Pharmaceutics-1628166)’. Please, see attached the main revisions and detailed responses to the referees’ comments are listed below. We have also made a number of further changes after carefully rereading the manuscript, which are likewise highlighted in red. 

Reviewer 2 Report

The review manuscript submitted by da Rocha et al. provides an overview on the biomedical/biotechnological applications of the clay materials; particularly focusing on the conservation of the photostability of the pharmaceutically active compounds by clay materials.

The review article compiles reports from, more or less, last 5 years; therefore it can be considered an up-to-date review article. However, looking at the extent of the manuscript, it can be considered as a "mini review" article. While I do not have serious concerns about the content and the style of the article, I can not be sure whether it would be significant contribution to the scientific community or not, since the manuscript does not involve any critical view points about the scientific topic but only reports compiled from the literature.

Although I do not have any serious concerns, I would like to highlight the following issues that should be taken under consideration by the Authors to improve the quality of the article:

  • The sentence in lines 57-58 about photo-toxicity must be grammatically restructured for clarity. 
  • Introduction section starts with photo-stability of the pharmaceutically active agents. Introduction to photo-stability problem is exemplified within lines 61-82 by providing information from literature. However, starting from line 83, the role of clay materials is presented in a very sudden manner. The transition from the paragraphs exemplifying the photo-stability issues to the paragraphs about the usability of clay materials for the given purpose sounds unrelated. In other words, the objective behind writing the review article - which was obviously the use of clay materials in pharmaceutical applications- somehow loses its importance. I recommend restructuring the Introduction section. In its current form, it does not sound convincing.
  • The following sentence starting from line 100 is unclear: "Its classification is the type of structure and chemical composition". I really cannot understand what the Authors meant about the classification being the type of structure. Please clarify this point.
  • It seems to me that sentence in lines 133-134 is grammatically wrong. Please pay attention.
  • the information provided within lines 138-139 sounds repeated unnecessarily.
  • The following sentence within lines 155-156 sounds incomplete: "Even though very little information is known about the photostability mechanism provided by clay minerals."
  • The following sentence within lines 156-158 sounds unclear: Knowing the importance of the various pharmaceutical or medical fields opens the possibility for new trends and challenges that address the role of clay minerals in the stability/protection of drugs". Please restructure the statement made in this sentence.
  • Once more, the information given within lines 235-237 about the role of clay materials in photostability sounds over-repeated.
  • The over-repeating problem about the photostability providing role of the clay materials also appears in lines 281 and 288. At this point of the article, it is already clear to the readers that the article is mainly about the photo-stability provided by clay materials. I think there is no need to repeat it too often. 
  • The information provided in lines 304-305 and 309-310 sounds repetitive, again.
  • The sentence in lines 319-321 reads: "These systems can be used in different science applications and therapeutic studies such as food technology, biotechnology, and the pharmaceutical industry." I suggest to limit it to biotechnology only, since the sentence appears under the subsection entitled "Trends in clays for biotechnological drugs stability". Claims should not be scattered too much.
  • The sentence within lines 381-383 sounds grammatically incorrect: "This review highlighted about stabilization of the different molecules onto clay minerals structures and future trends for drug-clay minerals hybrids in further promising studies."

Author Response

(The authors gave the same response as above.)

Reviewer 3 Report

The review entitled "Clays as vehicles for drug stability" provides an up-to-date overview of the possibility of using clay minerals in (photo) stabilization of drugs (including biotechnological) and cosmetic active ingredients. The paper is clearly and concisely written, and the structure of the paper is good. Given the content of the paper, the authors should consider whether the title of the paper “Clay minerals as vehicles for drug (photo) stability” is more appropriate.

However, references 27-45 are missing in the text, so it is not clear whether part of the text has been omitted.

Other, minor comments for the authors are included in the text.

Author Response

(The authors gave the same response as above.)
